# Economic freedom, inclusive growth, and financial development: A heterogeneous panel analysis of developing countries

**Zhengrong Yang[1]ᴼ, Prince Asare Vitenu-Sackey[2]ᴼ *, Lizhong Hao[3]‡, Yuqi Tao[4]‡**

**1** School of Finance and Business, Zhenjiang College, Zhenjiang, China, **2** Department of Economics, Strathclyde Business School, University of Strathclyde, Glasgow, United Kingdom, **3** Pamplin School of Business Administration, University of Portland, Portland, OR, United States of America, **4** Business School, University of Hong Kong, Hong Kong, China

ᴼ These authors contributed equally to this work.
‡ LH and YT also contributed equally to this work.
* prince.vitenu-sackey@strath.ac.uk, pavsackey@gmail.com

**Data Availability Statement:** The data that support this study can be found in the Mendeley Data repository at 10.17632/6bcpkytp52.1.

## Abstract

The effective and efficient management of financial systems and resources fosters a socio-economic climate conducive to technological and innovative advancement, thereby fostering long-term economic growth. The study used panel data from 72 countries classified as less financially developed between 2009 and 2017 to examine the role of economic freedom and inclusive growth in financial development. For the long-run estimations, we utilised the linear dynamic panel GMM-IV estimator, panel corrected standard errors (PCSE) linear regression method, and contemporaneous correlation estimator, a generalised least squares method. Our analyses indicate that economic liberty, inclusive growth, and capital stock significantly contribute to financial development in a positive manner. Moreover, inclusive growth contributes positively to overall financial development by enhancing economic freedom. Regardless of exogenous and endogenous shocks, we found that the tax burden and investment freedom are negative drivers of financial development as measured by the overall financial development index. In contrast, protection of property rights, government spending, monetary freedom, and financial freedom are positive and significant drivers of economic growth.

## Introduction

In the pursuit of sustainable economic development, finance is an important and relevant factor [1, 2]. However, countries with limited financial resources could be more productive if their financial resources and systems were managed effectively and efficiently [3]. Inadvertently, the effective and efficient management of financial systems and resources fosters a socioeconomic climate conducive to technological and innovation advancement, which fosters long-term economic growth [1–3]. Moreover, economic freedom creates two paths for growth: (i) the path for the development of new technologies and new designs, which advance technological

**Funding:** Z Y National Social Science Fund Project " Research on the Mechanism and Countermeasures of Digital Finance to Support SME Credit Community Financing" (No. 22BJY076). The funders had no role in study design, data collection and analysis, decision to publish, or preparation of the manuscript.

**Competing interests:** The authors have declared that no competing interests exist.

progress and serve as essential growth stimulants; and (ii) the level of market investment and openness of an economy. In other words, the function of legal structures, such as freedom from corruption, protection of property rights, effectiveness of the judiciary, etc., ensures the protection of the property rights of individuals and institutions [4]. The endogenous relationship between economic freedom, financial market crashes, and financial market structures has been established. As a result of economic freedom's unregulated framework, the probability of financial market collapses is inescapable. Nonetheless, economic liberty provides a degree of transparency that could reduce regulatory uncertainty and the likelihood of crashes [5]. Economic freedom is important for creating incentives. De Haan and Sturm [6] said that a country's growth or stagnation depends on its economic freedom or strong socioeconomic institutions.

According to finance-growth theory, the variation in the quality and quantity of financial systems is critical to the expansion of an economy. According to Fung [7] and Sadorsky [8], economic growth results from financial development through two channels: first, the effectiveness of financial systems leads to the accumulation of financial resources for productive use, and second, financial liberalisation promotes risk-sharing by increasing investment and decreasing the cost of equity, which results in economic growth. Given these factors, we can assert that financial development leads to inclusive growth via enhanced socioeconomic institutions or economic liberty. Individuals and businesses have the right to own property in a free economy, and minimal taxes ensure high participation in economic activity. Despite this, inclusive growth depends on stronger socioeconomic institutions (economic freedom) based on effective government, the rule of law, open markets, and effective regulation. Consequently, promoting economic freedom results in inclusive growth [9] and advancing financial development [8]. Contrary to the foregoing, increased taxes and tariffs, stringent regulatory controls, lax startup support, increased business investment, etc., do not provide more level playing fields and competitive markets—perhaps they are not pro-business policies that stimulate inclusive growth [10].

Numerous empirical studies have concluded that financial liberalisation, a subset of economic freedom that measures access to credit and capital markets, is strongly correlated with higher economic growth, less restrictive credit constraints, and lower consumption growth volatility for smaller firms [11, 12]. Other scholars, however, have demonstrated that financial liberalisation in capital and credit markets increases the efficiency of financial institutions, decreases intermediation costs, and improves economic outcomes through economic liberty [13, 14]. Kouton [15] analysed the connection between economic freedom and inclusive growth in Sub-Saharan Africa using a system GMM estimator from 1996 to 2016. The result indicates that economic freedom and inclusive growth have a positive and significant relationship, as they are highly interconnected. In support of this conclusion, Olayinka Kolawole [16] opined that the design and implementation of policies that could increase investment freedom, strengthen labour freedom, and ensure the protection of property rights would significantly contribute to inclusive growth. In other words, inclusive growth could result from job creation to generate a sustainable income through economic expansion. Coetzee and Kleynhans [17] recognise that economic freedom and economic growth are interdependent, as higher levels of economic growth result from greater economic freedom. Theoretically, according to Sergeyev [18], a higher level of financial development is likely to reduce the sensitivity of an economy. However, economic freedom as a result of financial development affects the susceptibility of growth to shocks.

Given these arguments, we attempt to delve deeply into the economic freedom-inclusive growth-financial development nexus in order to provide new evidence by: first employing the financial development index and its dimensions and sub-dimensions, thus financial markets and institution development indexes, as well as access, depth, and efficiency indexes. To the

best of our knowledge, no study has attempted to use these indexes to investigate this nexus. Most financial development studies employ proxies such as credit to the private sector, broad money to GDP, stock market capitalization, savings, loan growth rate, and so on [4, 18–22]. According to the IMF, numerous studies estimate the effect of financial development on economic growth, wealth distribution, and stability; typically, these studies use one or two variables for financial depth to represent financial development, such as stock market capitalization or domestic credit to the private sector (private credit to GDP). These measures, however, do not fully account for the complicated and multifaceted nature of financial development. As a result, we rely on this index to capture the complex, multifaceted nature of financial development, such as financial system depth, access, and efficiency.

Second, the individual effect on the dimensional indexes of financial development is assessed using the Heritage Foundation's twelve indicators of economic freedom as well as the overall economic freedom index. Because most studies rely on the overall index of economic freedom [15, 18], we tend to use the sub-dimensions to reveal their positive and negative effects, as well as the major drivers of financial development. Some studies, on the other hand, used the Fraiser Institute's overall index of economic freedom [5, 17, 20]. However, we use the Heritage Foundation's Economic Freedom Index with the assumption that economic freedom leads to increased prosperity, and that the Index of Economic Freedom demonstrates the positive relationship between economic freedom and a variety of beneficial social and economic goals. Economic freedom is strongly linked to better societies, healthier environments, higher per capita incomes, human progress, democracy, and poverty eradication.

Thirdly, we make use of the linear dynamic panel data GMM IV estimator [23], the panel corrected standard errors linear regression estimator, and the generalised least square with correlated disturbances (contemporaneous correlation) estimator. The linear dynamic panel data GMM-IV estimator fits dynamic models by employing either the Blundell-Bond/Arellano-Bover system estimator or the Arellano estimator to estimate complex models more easily than the estimators described previously [23]. For robust inference, we utilised the panel corrected standard errors (PCSE) linear regression and contemporaneous correlation estimator, and the generalised least square (GLS) with correlation disturbances methods. Using panel corrected standard errors (PCSE), we tend to resolve the issue of heteroscedasticity that could arise in the model's standard errors [24]–since the data series exhibited mixed order of integration, this method is essentially appropriate to ensure the explicit resolution of measurement errors. In addition, the PCSE could address serial correlation, autocorrelation, simultaneity bias, and heterogeneity. Consequently, we use the GLS with correlated disturbances estimator for robust confirmation of the PCSE's results. According to Koreisha and Fang [25], the GLS has the statistical advantage of identifying weak and inefficient parameters that can be estimated in the procedure and then corrected. In general, we employ these methods to address potential endogeneity concerns. For example, the GMM-IV method employs the lag effect of the dependent variable as a tool to control for any potential endogeneity bias.

Due to the financial crisis that occurred between 2007 and 2008, the study begins from 2009, with the assumption that, after a crisis, every country will tend to strengthen and improve the breadth, accessibility, and efficiency of its financial systems. According to De Haan and Sturm [26], in the aftermath of a crisis, countries tend to tighten regulations to allay fears of future uncertainties that could portend slower output growth–perhaps this strategy reduces economic freedom.

Our paper is organised as follows: Section 2 describes the theoretical context, Section 3 describes the econometric approach, including methodology, data, and variables, Section 4 presents the results and findings discussion, and Section 5 concludes the study.

## Theoretical background

We follow Sergeyev [18] theoretical proposition on the backdrop of Aghion et al. [20] theoretical model–-where the scholars contend that socioeconomic institutions and financial development can influence economic growth sensitivity to shocks. The model can be found below:

$$\Delta y_{i,t} = \alpha_0 + \alpha_1 . y_{i,t-1} + \alpha_2 . shock_{i,t} + \alpha_3 . credit_{i,t} + \gamma . \alpha_2 . shock_{i,t} . credit_{i,t} + \beta . X_{i,t} + \mu_i + \varepsilon_{i,t} \quad (1)$$

In Eq (1), $\Delta y_{i,t}$ and $y_{i,t-1}$ represents economic growth and its lagged of GDP per capita in logarithm, credit represents financial development, shock represents exogenous shocks thus annual inflation rate of oil prices multiplied by net fuel exports as a share of GDP, $\gamma$ represents output variability hence economic growth volatility, and $X_{i,t}$ represents the control variables such as the logarithm of government share in gross domestic product, investment ratio, and population growth rate.

The concept of investment composition is used to illustrate this mechanism in the explanation. Short-term investments face fewer obstacles than long-term investments because they do not necessitate a longer implementation period. These obstacles are essentially shocks, both endogenous and exogenous shocks, that affect an economy in various ways. Exogenous shocks influence an economy as a result of external interactions, particularly with the outside world, whereas endogenous shocks influence liquidity on occasion due to imperfections within the economy. In this context, weaker social and economic institutions would generate stronger shocks (economic freedom). Because property rights protection is weaker, agents are highly motivated to seize people's property. Moreover, raiders may inappropriately expropriate firm owners; consequently, the weaker the socioeconomic institutions, the more susceptible firms are to shocks. In other words, firms generate greater profits when the probability of overcoming shocks is greater. Profitability enables businesses to resist unjustified takeovers by employing competent attorneys to seek legal redress.

When firms recognise the risk of disruption from shocks, they are traditionally disincentivised from making long-term investments. Essentially, this occurs when there is a negative exogenous shock to productivity; when firms' profits decline, the probability of long-term investment interruptions increases. Unintentionally, exogenous shocks have a negative impact on investments, leading to weakened socioeconomic institutions (economic freedom). When the level of financial development is greater, firms are able to borrow; consequently, they have the capacity to withstand investment shocks and growth sensitivity. Improved socioeconomic institutions (economic liberty) pave the way for firms' and individuals' uniform access to financial institutions and markets [9].

Numerous scholars have demonstrated a correlation between socioeconomic institutions, economic growth, and the evolution of the financial system. These academics argue that economic institutions (quality of markets, protection of property rights, government integrity, etc.) and political institutions (media freedom, effectiveness of the judiciary, right to vote, etc.) represent economic freedom and ensure economic development by empowering the incentives of economic agents and distributing political authority [27, 28]. Beck and Levine [29] argue, in support of these scholarly works, that countries with robust legal structures ensure contract enforcement and property and investor rights protection, which strengthen financial systems and economic agents. In other words, efficient financial systems resulting from robust socioeconomic institutions result in the equitable distribution of capital among individuals and businesses, thereby ensuring economic efficiency, individual freedom, and social justice.

## Econometric approach

### Empirical strategy

To achieve the study's objective, we adopted some econometric approaches. The techniques used are (1) unit root test where we employed the tests of Pesaran [35] CIPS and CADF tests and Im, Pesaran and Shin [36] IPS test; (2) cross-sectional dependence test where we employed Pesaran [37] test; (3) cointegration test where we employed Pedroni [38], Westerlund [39] and Kao [40] cointegration tests; (4) variance inflation factor for multicollinearity test, homogeneity test [41], and correlation matrix where we used pairwise correlation test; (5) long-run parameter estimations where we used linear dynamic panel data GMM-IV estimator, panel corrected standard errors (PCSE) linear regression method, and contemporaneous correlation estimator thus generalized least square method.

First, we examine the data series for a unit root to determine whether the variables are stationary or nonstationary. Consequently, at a significance level of 5% or less, we expect to reject the unit root assumption. After determining that the variables are non-stationary, we conduct a cross-sectional dependence test to determine the cross-sectional independence of the individual error terms of the panels and a homogeneity test to determine the heterogeneity of the slope. The cointegration relationship between the dependent and independent variables is then examined. Evidence of a cointegration relationship suggests a long-run equilibrium between the dependent and independent variables; consequently, the estimations will depict the long-term relationships. In addition, a correlation matrix and variance inflation factor were computed to test for multicollinearity and the correlation signs of the variables. The multicollinearity rule of thumb states that no two or more independent variables should have correlation coefficients between -0.70 and +0.70 with the dependent variable [42]. The variance inflation factor value should be less than 10 and the tolerance level should be greater than 0.2. Multicollinearity evidence may result in erroneous long-term parameter coefficients.

After establishing a significant and dependable data series, we employ the linear dynamic panel data GMM-IV estimator to conduct long-run estimations. The linear dynamic panel data GMM-IV estimator is based on the Blundell and Bond [43] and Arellano and Bover [44] estimators, which use moment conditions in which the lagged levels of the predetermined and dependent variables serve as instruments for the differenced equation. In addition, the linear dynamic panel data GMM-IV estimator fits dynamic models by employing either the Blundell-Bond/Arellano-Bover system estimator or the Arellano estimator to estimate complex models more easily than those mentioned previously [23]. For robust inference, we utilised the panel corrected standard errors (PCSE) linear regression and contemporaneous correlation estimator and the generalised least square (GLS) with correlation disturbances method. Using panel corrected standard errors (PCSE), we tend to resolve the issue of heteroscedasticity that could arise in the model's standard errors [24]. In addition, the PCSE could address serial correlation, autocorrelation, simultaneity bias, and heterogeneity. Consequently, we use the GLS with correlated disturbances estimator for robust confirmation of the PCSE's results. According to Koreisha and Fang [25], the GLS has the statistical advantage of identifying weak and inefficient parameters that can be estimated in the procedure and then corrected. See Appendix in S1 File for more details about the methods.

## Empirical model

The econometric model proposed for our study can be found as follows:

$$\text{Financial development} = f(\text{economic freedom, inclusive growth, capital formation, population growth, exogenous shock, endogenous shock}) \quad (2)$$

After taking into account the natural logarithm of inclusive growth (GDP per person employed) and capital formation in Eq (2); the empirical models can be found as:

$$FDIX_{i,t} = \beta_0 + \beta_1 \, EFIO_{i,t} + \beta_2 \, LNGDPPC_{i,t} + \beta_3 \, GCF_{i,t} + \beta_4 \, POPG_{i,t} + \mu_{i,t} \quad (3)$$

$$FDIX_{i,t} = \beta_0 + \beta_1 \, EFIO_{i,t} + \beta_2 \, LNGDPPC_{i,t} + \beta_3 \, GCF_{i,t} + \beta_4 \, POPG_{i,t} \\ + \beta_5 \, EXOGSHOCK_{i,t} + \beta_6 \, ENDOSHOCK_{i,t} + \mu_{i,t} \quad (4)$$

In Eqs (3) and (4), *FDIX* denotes the overall financial development index. Financial development is measured by indexes of financial institutions' development and financial markets development that incorporate their level of access, depth, and efficiency. *EFIO* stands for economic freedom index and encompasses 12-dimensional indicators. *LNGDPPC* represents inclusive growth; thus, gross domestic product per person employed, *GCF* stands for gross capital formation, which denotes investment, and *POPG* stands for population growth. *EXOGSHOCK* and *ENDOSHOCK* represent exogenous and endogenous shocks, respectively: the real effective exchange rate and consumer price index volatility. *u* represents the error terms, *i* represents a cross-section or panel of 72 countries, and *t* represents the study period 2009 to 2017. The 12 indicators measuring economic freedom have been outlined in Table 1, and the other sub-indices of financial development–the sub-dimensions are considered in our proposed models. See S5 Table in S1 File for the list of countries used in the study.

## Data

The study used panel data on 72 countries classified as less financially developed from 2009 to 2017. The countries were selected based on the IMF's financial development index, with only those scoring below 0.5 being considered. However, any country below the financial development index's median value was classified as less financially developed. The dependent variable of our study is financial development with dimensions of financial institutions and markets development. However, the independent variables are economic freedom and inclusive growth. Economic freedom has 12 sub-dimensions: tax burden, government integrity, government spending, business freedom, investment freedom, financial freedom, labour freedom, monetary freedom, property right, judiciary effectiveness, tax burden, and trade freedom. Other variables such as population growth, gross capital formation, endogenous shock, and exogenous shock are used as control variables (see Table 1 for the description of variables).

We use Kouton's [15] study to measure inclusive growth using GDP per person employed, with support from Raheem and Isah [45]. Kouton [15] investigated the link between economic freedom and inclusive growth, whereas Raheem and Isah [45] investigated the link between natural resource rent, human capital development, and inclusive growth in Sub-Saharan Africa. We contend, based on their assumptions, that the conduit through which people can ultimately benefit from growth is through employment by earning income. In the same vein, we adopted GDP per person employed to measure inclusive growth by taking into account the economic benefits that may result from increased output rates, such as job creation, a reduction in abject poverty, and increased economic size. Furthermore, the United Nations uses the

**Table 1. Variables description and sources.**

| Indicator | Variable | Description | Source |
|---|---|---|---|
| FDIX | Financial development index | Measures the access, size, stability, and efficiency of the financial system on a score of 0 to 1 where 1 = strong and 0 = weak | IMF |
| FIIX | Financial institution development index | Measures the access, size, stability, and efficiency of financial institutions on a score of 0 to 1 where 1 = strong and 0 = weak | IMF |
| FMIX | Financial market development index | Measures the access, size, stability, and efficiency of financial markets on a score of 0 to 1 where 1 = strong and 0 = weak | IMF |
| FIDIX | Financial institution Depth index | Measures the size, and stability of financial institutions on a score of 0 to 1 where 1 = strong and 0 = weak | IMF |
| FIAIX | Financial institution Access index | Measures the access of financial institutions on a score of 0 to 1 where 1 = strong and 0 = weak | IMF |
| FIEIX | Financial institution Efficiency index | Measures the efficiency of financial institutions on a score of 0 to 1 where 1 = strong and 0 = weak | IMF |
| FMDIX | Financial Market Depth index | Measures the size and stability of financial markets on a score of 0 to 1 where 1 = strong and 0 = weak | IMF |
| FMAIX | Financial Market Access index | Measures the access of financial markets on a score of 0 to 1 where 1 = strong and 0 = weak | IMF |
| FMEIX | Financial Market Efficiency index | Measures the efficiency of financial markets on a score of 0 to 1 where 1 = strong and 0 = weak | IMF |
| efio | Economic freedom index | Measures the socioeconomic development of an economy and also institutional effectiveness. It encompasses 12 indicators and is measured on scores from 0 to 100, where 100 = total economic freedom and 0 = poor economic freedom | Heritage Foundation |
| efipr | Property right index | Measures the freedom of individuals to own private properties that are backed by laws of a country | Heritage Foundation |
| efije | Judiciary effectiveness | Measures the effectiveness of the judiciary function in the enforcement of the rule of law | Heritage Foundation |
| efigi | Government integrity | Measures governments effort to curb corruption for the avoidance of uncertainty and insecurity | Heritage Foundation |
| efitb | Tax burden | Measures the level of burden of taxes on individuals and businesses | Heritage Foundation |
| efigs | Government spending | Measures the extent of government expenditure | Heritage Foundation |
| efifh | fiscal health | Measures the extent of the burden on the government to provide fiscal projects | Heritage Foundation |
| efibf | Business freedom | Measures governments efficiency in regulating businesses | Heritage Foundation |
| efilf | Labour freedom | Measures the regulatory effectiveness of labour markets | Heritage Foundation |
| efimf | Monetary freedom | Measures price control effectiveness and price stability | Heritage Foundation |
| efitf | Trade freedom | Measures non-tariff and tariff impediments and effectiveness | Heritage Foundation |
| efiif | Investment freedom | Measures the restriction and flows of investment capital | Heritage Foundation |
| efiff | Financial freedom | Measures regulatory and efficiency of the banking sector | Heritage Foundation |
| lngdppc | Inclusive growth: | Gross domestic product per person employed US$ | World Development Indicators |
| lnpopg | Population growth rate | population growth % | World Development Indicators |
| lnGCF | Capital stock—Investment | Gross capital formation (constant 2010 US$) | World Development Indicators |
| exogshock | Exogenous shock | The annual standard deviation of the real effective exchange rate | Authors' computation |
| endoshock | Endogenous shock | The annual standard deviation of the consumer price index | Authors' computation |

annual growth rate of GDP per person employed to measure inclusive growth and Sustainable Development Goal #8. As a result, the United Nations urges governments and policymakers to prioritise the employment aspect of growth in order to significantly increase inclusive growth [15]. It is worth noting that inclusive growth promotes decent employment, which leads to income security in the long run.

Because unemployment harms the economy and ordinary citizens, inclusive growth [46] and economic freedom [47] result in productive employment. Furthermore, economic growth is dependent on financial development in a country with a higher level of financial

development. Such a country has a higher per capita income and economic growth [19, 20, 48]. Furthermore, regulatory reductions that result in financial liberalisation stimulate economic growth and reduce the likelihood of a financial crisis [49]. We tend to acknowledge endogenous and exogenous factors that could arise from productivity-enhancing investment volatility by including endogenous and exogenous shocks because robust financial constraints make growth and investment more vulnerable to shocks. In the same vein, the relationship between growth and volatility is inversely related [20]. To represent endogenous and exogenous shocks, we used the annual standard deviation of the consumer price index and the real effective exchange rate, respectively. Notably, volatility can be caused by a variety of factors, including changes in energy supply prices, government budget policies, exchange rate volatility, inflation volatility, and so on [20]. Furthermore, high volatility may exacerbate financial development because the ability to withstand liquidity shocks cannot be supported by weak financial markets and institutions [50]. Furthermore, we include population growth against the backdrop of Barro and Lee [51] policy variables to measure the effect of social policy with support from Blau [5].

## Results and discussion

### Results

**Summary statistics.** Table 2 displays the summary statistics for the variables in the study. Given the countries' heterogeneous economic characteristics, we found that the average index score for financial development was 0.296 with a standard deviation of 0.140 annually. Whereas the economic freedom index score averaged 4.019 per year with a standard deviation of 0.145, the GDP per person employed increased by 9.114% per year with a standard deviation of 0.862%. Furthermore, gross capital formation, a measure of capital investment, increased at an annual rate of 22.79% on average. This implies that countries in our sample invested in more capital projects during the sample period, thereby supporting financial system growth and promoting economic freedom and inclusive growth. The population growth rate, on the

**Table 2. Descriptive statistics.**

|  | FDIX | FIAIX | FIDIX | FIEIX | FIIX | FMAIX | FMDIX | FMEIX | FMIX |
|---|---|---|---|---|---|---|---|---|---|
| Mean | 0.296 | 0.372 | 0.233 | 0.627 | 0.422 | 0.159 | 0.180 | 0.153 | 0.164 |
| Std. Dev. | 0.140 | 0.225 | 0.175 | 0.146 | 0.150 | 0.208 | 0.181 | 0.237 | 0.173 |
| Jarque-Bera | 66.279 | 30.623 | 671.511 | 52.981 | 13.909 | 123.614 | 516.920 | 1078.729 | 190.399 |
| Probability | 0.000 | 0.000 | 0.000 | 0.000 | 0.001 | 0.000 | 0.000 | 0.000 | 0.000 |
| Observations | 648 | 648 | 648 | 648 | 648 | 648 | 648 | 648 | 648 |
|  | EFIPR | EFITB | EFITF | EFIBF | EFIFF | EFIFH | EFIGI | EFIGS | EFIO |
| Mean | 3.551 | 4.380 | 4.332 | 4.152 | 3.862 | 0.454 | 3.511 | 4.192 | 4.091 |
| Std. Dev. | 0.540 | 0.088 | 0.132 | 0.201 | 0.386 | 1.311 | 0.385 | 0.387 | 0.145 |
| Jarque-Bera | 585.838 | 33.757 | 325.940 | 54.647 | 548.361 | 1356.400 | 3350.574 | 49282.120 | 228.513 |
| Probability | 0.000 | 0.000 | 0.000 | 0.000 | 0.000 | 0.000 | 0.000 | 0.000 | 0.000 |
| Observations | 648 | 648 | 648 | 648 | 648 | 648 | 648 | 648 | 648 |
|  | EFILF | EFIMF | LNGDPPC | LNGCF | LNPOPG | EFIIF | EFIJE |  |  |
| Mean | 4.069 | 4.290 | 9.114 | 22.799 | 0.023 | 3.835 | 0.409 |  |  |
| Std. Dev. | 0.270 | 0.122 | 0.862 | 4.439 | 0.938 | 0.727 | 1.165 |  |  |
| Jarque-Bera | 114.513 | 35505.670 | 107.114 | 11213.630 | 548.862 | 4916.905 | 1245.980 |  |  |
| Probability | 0.000 | 0.000 | 0.000 | 0.000 | 0.000 | 0.000 | 0.000 |  |  |
| Observations | 648 | 648 | 648 | 648 | 648 | 648 | 648 |  |  |

other hand, was 0.023% per year on average, with a standard deviation of 0.938%. (See Table 3 for more details.) More importantly, the Jarque-Bera test, which shows p-values less than 0.05, confirmed that our data series is not normally distributed.

**Panel unit root and cross-sectional dependence tests.** We used unit root tests to determine the degree of stationarity of the variables in the study. Table 4 does, however, show the results of the tests. Unit root tests were performed on Pesaran [35] and Im, Pesaran and Shin [36]. According to the results, there is no unit root in the variables because IPS tests rejected the unit root assumption at the first difference. Meanwhile, CIPS and CADF depicted mixed order of integration. Despite the inconsistent results of the tests, we find evidence that the variables are stationary at first difference at 1% and 5% significance levels using the first generation unit root test, thus IPS. In addition, we used the Pesaran [37] cross-sectional dependence test, which has statistical power to detect any weak cross-sectional dependency. According to the results, all variables have cross-sectional dependence; thus, the error terms of the variables correlate in the individual panels. We present the cross-sectional dependence test results in S6 Table in S1 File.

**Cointegration tests.** Checking for long-term relationships or equilibrium between dependent and independent variables is essential because it provides confidence in the estimation of long-term parameters. In this regard, three cointegration tests were conducted to determine the long-term relationship between the dependent and independent variables of the study. Table 5 presents the results. The results of the tests indicate that the variables are cointegrated, as they have a long-run relationship. Specifically, tests conducted by Kao [40] and Westerlund [39]confirmed cointegration at 1% and 5% significance levels. In addition, the Pedroni [38] test confirmed, at a 1% significance level, that the variables have a long-term relationship within and between dimensions.

**Correlation, multicollinearity and homogeneity tests.** Table 6 displays the results of our correlation matrix; for the sake of clarity, we present only the results of the table's primary variables. According to the results, however, there was no evidence of multicollinearity because none of the independent variables exhibited a strong correlation with the dependent variables. Specifically, only LNGDPPC had a high correlation coefficient, but it did not meet the criteria for collinearity. According to Sun and Tong [42], no two or more independent variables should have correlation coefficients with dependent variables greater than—or + 0.70. On the other hand, we found significant positive correlations between economic freedom, inclusive growth, gross capital formation, and financial development, whereas population growth demonstrated a significant negative correlation.

The assumption of multicollinearity between the dependent and independent variables is rejected based on the results of the multicollinearity test presented in Table 6. Specifically, the VIF values of the variables were all less than 10, and the tolerance levels were also greater than 0. The correlation coefficients did not reveal any collinearity or multicollinearity. In contrast, Table 6 displays the homogeneity test performed to determine whether the slope coefficients of the parameters to be estimated are heterogeneous. The assumption that the slope is homogeneous is rejected at the 1% and 5% significance levels when the delta and adj. delta values of the homogeneity test are considered.

**Heterogeneous analysis of economic freedom, inclusive growth, and financial development.** After confirming cointegration and conducting satisfactory pre-diagnostic tests, long-run estimations between the dependent and independent variables were conducted. In this regard, we utilised the linear dynamic panel data GMM-IV estimator, which has the ability to resolve the model's potential reverse causality and endogeneity. Based on our findings (see Table 7), we observed a positive and significant correlation between economic freedom and overall financial development, as well as inclusive growth and overall financial development.

**Table 3. Relevant and related literature review.**

| Author(s) | Methodology, Sample & Context | Findings |
|---|---|---|
| D'Agostino et al. [30] | • Panel study using instrumental variable method (2SLS)<br>• Coverage: 152 countries<br>• Period: 1995 to 2017<br>• Topic: "**Does the economic freedom hinder the underground economy? Evidence from a cross-country analysis**" | • They conclude that the shadow economy suffers when there is a change in the economic freedom index because of state deregulation of financial markets.<br>• In addition, they illustrate the U-shaped relationship between the composite indicator of economic freedom and the shadow economy, which is supported exclusively by legal system freedom, business regulation, and property rights. |
| Ofoeda et al. [31] | • Panel study using system GMM with two step approach<br>• Coverage: 52 African countries<br>• Period: 2002–2019<br>• Topic: "**Financial inclusion and economic growth: What roles do institutions and financial regulation play?**" | • Their findings indicate that institutional quality augments the effect of financial inclusion on economic growth, whereas financial inclusion has profound and significant effects on institutional quality, financial regulation, and all surrogate measures of institutional quality. Moreover, financial regulation mitigates the effect of financial inclusion on economic growth. |
| Islam & Alhamad [32] | • Panel study of topmost remittance-earning economies using PMG<br>• Coverage: 10 countries<br>• Period: 1996 to 2019<br>• Topic: "**Impact of financial development and institutional quality on remittance-growth nexus: evidence from the topmost remittance-earning economies**" | • The results indicate that personal remittances have an irregular effect on economic growth; their positive spillovers have a negative effect, while their negative shocks have a positive effect. Both financial development and institutional quality have a positive effect on economic growth over the long term. There is no evidence that financial growth has a threshold effect on economic growth. |
| Van et al. [33] | • Panel study using advanced panel smooth transition regression method<br>• Coverage: 19 countries<br>• Topic: "**The asymmetric effects of institutional quality on financial inclusion in the Asia-pacific region**" | • They conclude that different income brackets experience different effects of institutional quality on financial inclusion. They conclude that middle-income countries like Vietnam and other emerging nations in the Asia-Pacific region benefit greatly from institutional reform, which is essential for ensuring prospective inclusive economic growth. |
| Aluko & Ibrahim [34] | • Panel study using threshold instrumental variable method<br>• Coverage: 28 Sub-Saharan African countries<br>• Period: 1996 to 2015<br>• Topic: "**Institutions and the financial development–economic growth nexus in sub-Saharan Africa**" | • The authors conclude that financial development promotes economic growth. The effect on growth is disproportionately strong given the quality of institutions. When the ICRG-based measure of institutional quality is used as the threshold variable, their findings show that financial development does not support economic growth significantly below the optimal level of institutional quality. Higher levels of finance are associated with economic expansion in countries where institutional quality exceeds a certain threshold.<br>• Financial development is found to have a significant impact regardless of whether a country is below or above the threshold when institutions are measured using the World Governance Indicators (WGI) proxy. It is worth noting that countries with fewer institutions benefit more from finance's ability to stimulate economic growth. Although it may appear counterintuitive, we believe that the relatively small impact of finance in countries with strong institutions indicates that these countries have reached a point of institutional saturation, where further advancements in the overall financial sector, while beneficial, have little impact on economic growth. |
| Kouton [15] | • Panel study using system GMM<br>• Coverage: 30 Sub-Saharan African countries<br>• Period: 1996–2016<br>• Topic: "**Relationship between economic freedom and inclusive growth: a dynamic panel analysis for sub-Saharan African countries**" | • Their findings suggest that the degree of economic freedom and changes in the degree of economic freedom have a positive and substantial effect on inclusive growth. It is demonstrated that economic freedom and inclusive growth are causally related, but not vice versa. |

The findings indicate that regardless of the existence of shocks, an increase in economic freedom and the promotion of inclusive growth could substantially boost a nation's financial development. Additionally, economic freedom and inclusive growth have a positive relationship with the development of financial institutions and markets. In S1 Table in S1 File, we present the results of our analyses pertaining to the disaggregate indexes of financial development, i.e., the indexes for financial institutions and markets.

In an effort to determine the indirect effect of economic freedom and inclusive growth on the sub-dimensions of financial institutions and market development as a composite measure

**Table 4. Unit root tests.**

|  | CIPS | | CADF | | IPS | |
|---|---|---|---|---|---|---|
|  | Level | First Diff. | Level | First Diff. | Level | First Diff. |
| FDIX | -2.785*** | -2.864*** | -5.232*** | 27.027 | -4.209*** | -14.843*** |
| FIIX | -2.464** | -2.752*** | -2.382** | 27.027 | -1.816** | -12.662*** |
| FMIX | -2.756*** | -3.066*** | -5.337*** | 27.027 | -17.910*** | -42.401*** |
| FIDIX | -2.100 | -2.495** | -5.464*** | 27.027 | -1.790** | -12.456*** |
| FIAIX | -2.182 | -2.766*** | -5.874*** | 27.027 | -6.142*** | -12.347*** |
| FIEIX | -2.349** | -2.927*** | -4.114*** | 27.027 | -1.552** | -12.568*** |
| FMDIX | -2.025 | -2.670*** | -3.502*** | 27.027 | -13.527*** | -11.586*** |
| FMAIX | -0.873 | -1.636 | 6.148 | 27.027 | -1.277** | -7.070*** |
| FMEIX | -2.183 | -2.114 | -8.381*** | 27.027 | -2383.57*** | -2927.43*** |
| efio | -2.024 | -2.598*** | 1.842 | 27.027 | 0.090** | -8.080*** |
| efipr | -1.307 | -1.565 | 9.596 | 26.748 | -3.161*** | -4.606*** |
| efije | -2.610*** | -2.610*** | 27.027 | 27.027 | 1.569 | 6.352** |
| efigi | -1.490 | -2.072 | -2.556** | 27.027 | 1.751 | -7.134*** |
| efitb | -2.771*** | -3.081*** | -4.539*** | 27.027 | -16.517*** | -13.816*** |
| efigs | -1.961 | -2.541*** | -0.416 | 27.027 | -3.101*** | -10.371*** |
| efifh | 2.610*** | 2.610*** | 27.027 | 27.027 | 2.254 | 1.524** |
| efibf | -1.608 | -2.493** | 4.104 | 27.027 | -0.648 | -8.263*** |
| efilf | -2.465** | -2.950*** | -6.126*** | 27.027 | -2.967** | -11.121*** |
| efimf | -1.471 | -2.222** | -1.825** | 27.027 | -2.123** | -12.019*** |
| efitf | -2.315** | -2.650*** | 1.151 | 27.207 | -8.862*** | -14.437*** |
| efiif | -2.004 | -2.365** | 2.314 | 27.027 | -5.708*** | -11.029*** |
| efiff | 1.646 | 1.202 | 19.972 | 25.491 | 1.053 | -2.718** |
| lngdppc | -1.151 | -2.591*** | -4.212*** | 27.027 | 0.401 | -7.626*** |
| lnpopg | -0.696 | -0.660 | -1.866** | 27.027 | -14.061*** | -13.754*** |
| lnGCF | -1.941 | -2.252** | -1.014 | 27.027 | -13.463*** | -9.431*** |
| exogshock | -2.068 | -0.011 | 10.135 | 27.027 | 6.206 | -13.184*** |
| endoshock | -0.314 | -0.777 | 4.965 | 27.027 | -33.683*** | 4.078** |

Note:

*** denote 1% significance level

** denote 5% significance level. CIPS and CADF = Pesaran (2007) test, IPS = Im et al. (2003) test.

of overall financial development, we investigated access, depth, and efficiency. Economic free-dom has a positive relationship with the efficiency of financial institutions and the access, depth, and efficiency of financial markets, except for the access of financial institutions, which is negative.

On the other hand, we found that inclusive growth contributes positively to the access, depth, and efficiency of financial institutions and the depth of financial markets, but negatively to their access and efficiency. Invariably, the models in which the relationships between eco-nomic freedom, inclusive growth, and sub-dimensions of financial institutions and markets were inverse suffered from autocorrelation, indicating a statistical inability to infer the results. For a robust inference, we used the linear panel corrected standard errors (PSCE) method and the generalised least squares (GLS) estimator to resolve the issue of autocorrelation and any correlation disturbances encountered by the linear dynamic panel data GMM-IV estimator during the analyses. The results of the robustness test are shown in S2-S4 Tables in S1 File. In the estimations, we relied on two models, model 1 of which does not account for shocks (both

**Table 5. Cointegration tests.**

| Panel cointegration test | | | | | | |
|---|---|---|---|---|---|---|
| **Kao Residual Cointegration Test** | | | | | | |
| | t-Statistic | Prob. | Sig. | | | |
| ADF | -2.395 | 0.008 | ** | | | |
| **Westerlund Cointegration test** | | | | | | |
| | statistic | Prob. | Sig. | | | |
| Variance ratio | 5.365 | 0.000 | *** | | | |
| **Pedroni Residual Cointegration Test** | | | Weighted | | | |
| | Statistic | Prob. | Sig. | Statistic | Prob. | Sig. |
| Panel v-Statistic | -3.384 | 1.000 | | -6.607 | 1.000 | |
| Panel rho-Statistic | 6.549 | 1.000 | | 6.977 | 1.000 | |
| Panel PP-Statistic | -11.603 | 0.000 | *** | -16.954 | 0.000 | *** |
| Panel ADF-Statistic | -7.446 | 0.000 | *** | -7.337 | 0.000 | *** |
| Alternative hypothesis: individual AR coefs. (between-dimension) | | | | | | |
| | Statistic | Prob. | | | | |
| Group rho-Statistic | 10.347 | 1.000 | | | | |
| Group PP-Statistic | -25.188 | 0.000 | *** | | | |
| Group ADF-Statistic | -9.175 | 0.000 | *** | | | |

Note:

*** denote 1% significance level

** denote 5% significance level. Only the cointegration tests of the main model are reported for simplicity sake.

**Table 6. Correlation matrix.**

| Correlation | | | | | |
|---|---|---|---|---|---|
| Probability | FDIX | EFIO | LNGDPPC | LNGCF | LNPOPG |
| FDIX | 1 | | | | |
| efio | 0.288*** | 1 | | | |
| lngdppc | 0.619*** | 0.443*** | 1 | | |
| lngcf | 0.310*** | -0.035 | 0.265*** | 1 | |
| lnpopg | -0.352*** | -0.323*** | -0.517*** | -0.160*** | 1 |
| Multicollinearity test | VIF | Tolerance (1/VIF) | | | |
| FDIX | - | - | | | |
| efio | 1.39 | 0.721 | | | |
| lngdppc | 1.68 | 0.594 | | | |
| lngcf | 1.58 | 0.632 | | | |
| lnpopg | 1.41 | 0.710 | | | |
| endoshock | 1.15 | 0.873 | | | |
| exogshock | 1.43 | 0.701 | | | |
| Test for slope heterogeneity—Pesaran and Yamagata homogeneity test | | | | | |
| Test | Value | Prob. | | | |
| Delta | 2.496** | 0.013 | | | |
| Adj. Delta. | 7.487*** | 0.000 | | | |

Note:

*** denote 1% significance level

** denote 5% significance level. Only the main model's correlation, homogeneity, and multicollinearity tests are reported for simplicity sake.

**Table 7. Linear dynamic panel data instrumental variables GMM estimation results of economic freedom, inclusive growth, and financial development.**

|  | DPD-GMM-1 | DPD-GMM-2 | DPD-GMM-1 | DPD-GMM-2 | DPD-GMM-1 | DPD-GMM-2 |
|---|---|---|---|---|---|---|
|  | FDIX | FDIX | FIIX | FIIX | FMIX | FMIX |
| efio | 0.037 | 0.039 | 0.041 | 0.009 | 0.037 | 0.062 |
|  | (17.76)*** | (12.02)*** | (16.78)*** | (2.17)** | (10.18)*** | (16.82)*** |
| lngdppc | 0.093 | 0.088 | 0.226 | 0.211 | 0.043 | 0.028 |
|  | (110.01)*** | (69.28)*** | (76.48)*** | (57.60)*** | (17.45)*** | (21.33)*** |
| lnpopg | -0.002 | -0.002 | -0.0004 | -0.001 | -0.001 | -0.002 |
|  | (-37.07)*** | (-27.01)*** | (-0.72) | (-0.75)*** | (-31.99)*** | (-48.68)*** |
| lnGCF | 0.011 | 0.011 | -0.001 | -0.002 | 0.022 | 0.017 |
|  | (9.78)*** | (8.26)*** | (-1.05) | (-0.75) | (21.37)*** | (16.98)*** |
| exogshock |  | -0.006 |  | -0.010 |  | -0.001 |
|  |  | (-11.55)*** |  | (-15.43)*** |  | (-4.42)*** |
| endoshock |  | 0.135 |  | 0.666 |  | -0.358 |
|  |  | (5.90)*** |  | (13.75)*** |  | (-27.41)*** |
| constant | -0.960 | -1.057 | -1.780 | -2.281 | -0.107 | 0.222 |
|  | (-42.19)*** | (-28.22)*** | (-61.34)*** | (-32.96)*** | (-6.55)*** | (6.94)*** |
| Wald chi2 | 32694.00*** | 16495.68*** | 8174.82*** | 5447.36*** | 29961.10*** | 44617.44*** |
| Sargan test | 64.715(1.000) | 63.382(1.000) | 70.864(1.000) | 65.176(1.000) | 67.548(1.000) | 67.550(1.000) |
| Autocorrelation: |  |  |  |  |  |  |
| ART (1) | 0.914(0.361) | 0.890(0.374) | -0.032(0.975) | -0.130(0.896) | 0.897(0.370) | 0.817(0.414) |
| ART (2) | -0.621(0.535) | -0.646(0.518) | -0.896(0.370) | -0.893(0.372) | -0.227(0.821) | -0.213(0.831) |
| No of Instruments | 141 | 180 | 141 | 180 | 141 | 180 |
| observation | 648 | 648 | 648 | 648 | 648 | 648 |

Note:

*** denote 1% significance level

** denote 5% significance level. Z-statistics are in parentheses. 1 = model without exogenous and endogenous shocks, 2 = model with exogenous and endogenous shocks. ART(1) and (2) as well as Sargan tests p-values are in parenthese

endogenous and exogenous), while model 2 does. In addition, S1 Table in S1 File details the estimations that considered the overall index of financial development and economic freedom, the disaggregate index of financial development, i.e., the financial market development index, and the financial institution development index, whereas S3 Table in S1 File details the estimations of the sub-dimensions of the financial institutions and financial markets development indices, i.e., depth, access, and efficiency. The disaggregated index of economic freedom and its association with financial development (global index, disaggregated index, and sub-dimensions) and inclusive growth are presented in S3 Table in S1 File.

Our estimations indicate that economic freedom, inclusive growth, and capital stock (gross capital formation) contribute positively and significantly to financial development. Specifically, we found that a one-percentage-point increase in overall economic freedom can boost financial development by 0.043%, 0.040%, and 0.033% at the 1% and 5% significance levels, even in the presence of negative endogenous and exogenous shocks. Moreover, inclusive growth positively contributes to overall financial development via enhanced economic freedom, such that a percentage point increase in inclusive growth could result in a 0.087%, 0.088%, or 0.089% increase in financial development at a 1% significance level. In an account of capital stock, there is evidence that it is essential and significant in terms of financial development—a percentage point increase in gross capital formation (capital stock) could significantly lead to a 0.005% to 0.006% increase in financial development at a 1% significance level. Despite the presence of exogenous

or endogenous shocks, we discovered that the population growth rate has a negative impact on economic growth. Specifically, a percentage point increase in population growth rate could negate financial development by 0.005% to 0.006% at a 5% significance level; meanwhile, our robust estimation yielded a coefficient that was insignificant.

Taking into account the disaggregate or dimensional financial development index measures—the financial institutions development index and the financial markets development index—we found that economic freedom and inclusive growth positively impact the development of financial markets and institutions. However, gross capital formation (capital stock) and population growth are inconsistent. In the event of shocks, endogenous and exogenous shocks have negative and positive effects on the capital stock, respectively, and contribute insignificantly to the development of financial institutions. In the absence of shocks, however, capital stock (gross capital formation) positively influences the relationship between economic freedom, inclusive growth, and financial development. Meanwhile, regardless of the presence of shocks, capital stock contributes positively to the development of financial markets. On the other hand, population growth contributes negatively and significantly to the development of financial institutions but plays no role in the development of financial markets.

In addition, we analysed the influence of economic freedom and inclusive growth on the sub-dimensions of financial institutions and market development measures (see S3 Table in S1 File). In particular, we considered the access, depth, and efficiency indexes of financial institutions as well as the access, depth, and efficiency indexes of financial markets. Regarding financial institutions, we found that economic freedom contributes positively and significantly to depth and efficiency but negatively to accessibility, regardless of the presence of shocks. And inclusive growth contributes positively and significantly to the accessibility, depth, and efficiency of financial institutions. In an account of capital stock, we observed a positive relationship between the efficiency of financial institutions and the depth of financial institutions when neither endogenous nor exogenous shocks are present. Meanwhile, we observed a negative and significant impact of capital stock on the access of financial institutions in the presence of exogenous and endogenous shocks but no impact in the absence of shocks. Moreover, population growth positively influences the relationship between economic freedom, inclusive growth, and the depth of financial institutions but negatively influences the access and efficiency of financial institutions. In terms of the development of financial markets, we observed a positive and statistically significant relationship between economic freedom and access to and depth of financial markets.

In contrast, the relationship between economic freedom and the efficiency of financial markets was negative and statistically significant, regardless of the presence of shocks. However, we observed a significant positive effect of capital stock (gross capital formation) on the accessibility, depth, and efficiency of financial markets, regardless of the presence of shocks. On the other hand, we observed a positive and significant effect of population growth on the access and depth of financial markets but a negative effect on their efficiency, regardless of exogenous or endogenous shocks. The results of our estimations of the sub-dimensions of economic freedom and financial development are presented in S4 Table in S1 File. Prior to that, we conducted analyses with the overall financial development index and economic freedom subdimensions in mind. Regardless of exogenous and endogenous shocks, the assessments found that tax burden and investment freedom are negative drivers of financial development as measured by the overall financial development index. In contrast, we found that the positive and significant drivers of financial development are the protection of property rights, government spending, monetary freedom, and financial freedom. Moreover, inclusive growth intervenes proportionally in this relationship; nonetheless, inclusive growth is a significant factor for economic development. We observed that the positive drivers of the development of

financial institutions are business freedom, trade freedom, government integrity, monetary freedom, and financial freedom.

Investment freedom and tax burden, on the other hand, have the opposite effect on the development of financial institutions. In an account of the development of financial markets, we observed that positive drivers include protection of property rights, government spending, monetary freedom, and financial freedom. On the other hand, what hurts the growth of financial markets are free trade, freedom to invest, and high taxes. In addition, we investigated the sub-dimensions of financial markets and institutions—namely, access, depth, and efficiency—to decipher the impact of economic freedom on them. Tax burden and investment freedom are identical factors that consistently and negatively affect the accessibility, depth, and efficiency of financial institutions and markets. On the other hand, financial freedom makes financial institutions and markets more accessible, deeper, and better at what they do (see S4 Table in S1 File). Also, Fig 1 depicts the overall impact of economic freedom on financial development with influence of inclusive growth.

## Discussion

To determine whether our data series is statistically reliable for estimations, preliminary tests were conducted. Validity and reliability were confirmed. Specifically, we found no evidence of unit root, multicollinearity, cointegration, and cross-sectional dependence. Our findings

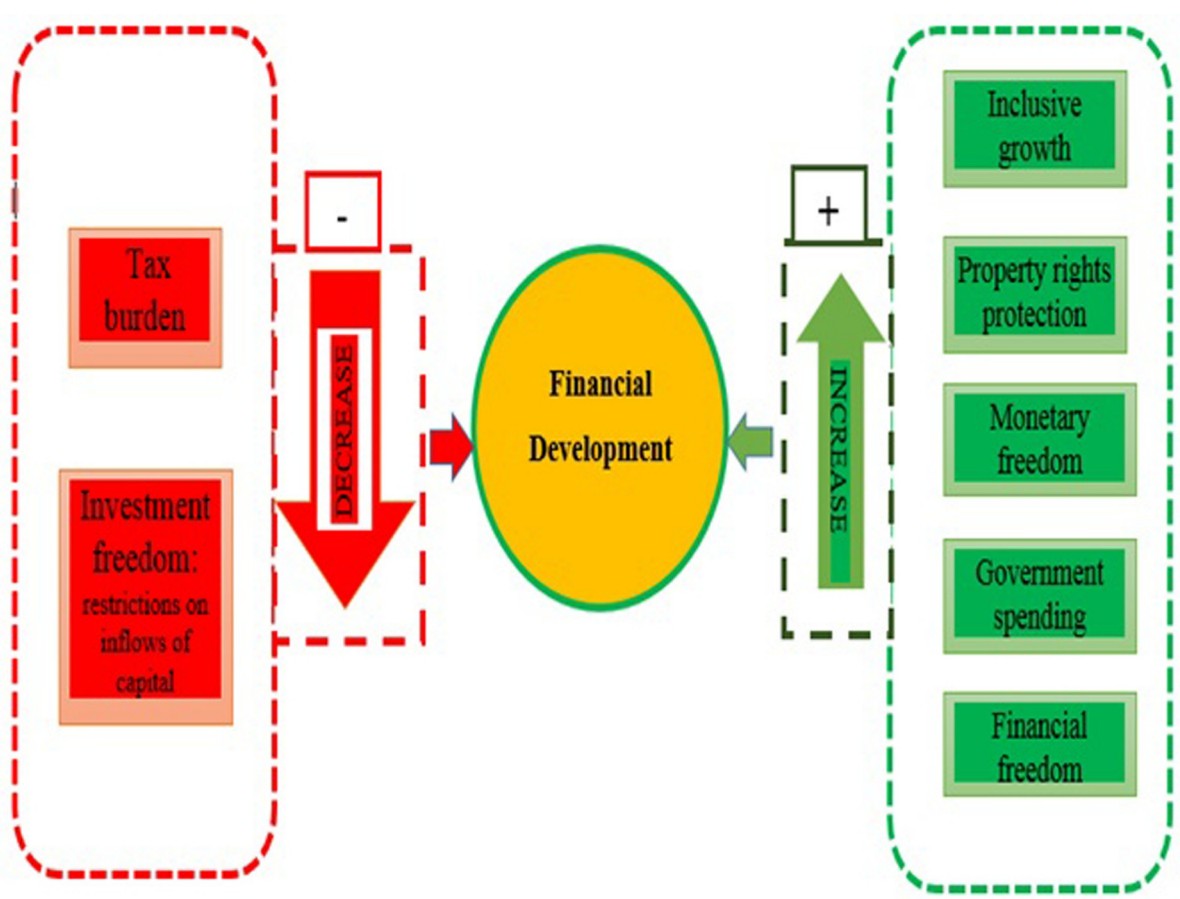

**Fig 1. Pictorial display of the significant drivers of financial development.**

suggest that economic freedom that reflects the development of socioeconomic institutions influences financial development positively through inclusive growth, i.e., GDP per employed person. This finding backs up the claims of Assi et al. [4], Isiksal et al. [3], Kouton [15], Murray and Press [9], and Sadorsky [8], who argue that governments' effectiveness in ensuring the independence of economic and political institutions, property rights protection, and tax reduction ensure inclusive growth—as well as strengthening financial sector development in a sustainable manner. To support this claim, Blau [5] exegetically confirmed that the strength of property rights protection and the level of free trade contribute to a lesser extent to the reduction of financial market crashes. In addition, the level of transparency in an economy as a result of economic freedom mitigates the uncertainties surrounding regulation and likely reduces the likelihood of shocks and crashes.

When an economy's level of financial development is high, its sensitivity decreases [18]. In addition, Sergeyev [18] argues that economic freedom (socioeconomic institutions) affects the sensitivity of economic growth to endogenous and exogenous shocks. However, a greater level of financial development mitigates the severity of shocks. Nonetheless, it is essential to strengthen the legal framework and policy initiatives designed to effectively and efficiently manage the financial system or sector to withstand fluctuations or uncertainties [3]—this supports our findings that financial and monetary freedom is essential and positively contributes to financial development. In nations with a high degree of economic freedom, financial development ensures financial resources accumulation and financial liberalisation that seeks to put unproductive resources to beneficial use, boosts investment [10] and reduces equity costs [4, 18, 20], which ultimately results in inclusive economic growth.

## Conclusion and practical implication

### Conclusion

The study used panel data on 72 countries classified as less financially developed from 2009 to 2017 to critically assess the role of economic freedom and inclusive growth in financial development. To achieve the study's objective, we adopted some econometric methodologies. The methodologies used are (i) unit root test where we employed the tests of Pesaran [35] CIPS and CADF tests and Im, Pesaran & Shin [36] IPS test; (ii) cross-sectional dependence test where we employed Pesaran [37] test; (iii) cointegration test where we employed Westerlund [39], Pedroni [38] and Kao [40] cointegration tests; (iv) correlation matrix where we used pairwise correlation test; Pesaran and Yamagata homogeneity test, and variance inflation factor, (v) long-run parameter estimations where we used linear dynamic panel data GMM-IV estimator, panel corrected standard errors (PCSE) linear regression method, and contemporaneous correlation estimator thus generalized least square method.

Our estimations indicate that economic freedom, inclusive growth, and capital stock positively contribute significantly to financial development. Specifically, we found that a one-percentage-point increase in overall economic freedom can boost financial development by 0.043%, 0.040%, and 0.033% at the 1% and 5% significance levels, even in the presence of negative endogenous and exogenous shocks. Moreover, inclusive growth positively contributes to overall financial development via enhanced economic freedom, such that a percentage point increase in inclusive growth could result in a 0.087%, 0.088%, or 0.089% increase in financial development at a 1% significance level. Regardless of exogenous and endogenous shocks, we found that tax burden and investment freedom are negative drivers of financial development as measured by the overall financial development index. In contrast, the protection of property rights, government spending, monetary liberty, and financial liberty are positive and significant drivers of economic growth in support of D'Agostino et al. [30]. According to the study's

findings, improved and strengthened financial sector regulatory and efficiency leads to financial development but reduces investment freedom; that is, restrictions on capital inflows and a high level of tax burden hinder or impede financial institutions' and markets' access, depth, and efficiency. Moreover, the development of the financial sector depends on monetary freedom, which reflects price stability and effective price control, and business freedom (business regulatory effectiveness and efficiency). Cross [10] elaborated, in support of these findings, that competitive markets and level playing fields for businesses are the overarching pro-business policies resulting from tax reduction, regulatory control limitation, startup empowerment, reining in occupational licensure rules, boosting business investment, and tariff elimination which is in support of Ding et al. [52], Huang J, Ulanowicz [53] and Singhal [54]. Our findings are consistent with the fiance-growth theory, which posits that variation in the quality and quantity of financial systems is crucial to the expansion of an economy, and that the efficiency of financial systems may lead to the accumulation of financial resources for productive use. In addition, financial liberalisation encourages risk-sharing by boosting investment and reducing the cost of equity, resulting in economic expansion. Therefore, financial development facilitates inclusive growth through improved socioeconomic institutions or economic liberty.

## Practical implication

Our findings imply that property rights protection regulations and laws should be effectively guarded and enforced due to their contribution to economic growth. Importantly, financial sector regulations and rules should be staffed effectively and efficiently to ensure the financial sector's continued growth. In addition, price stability and price control mechanisms should be designed and strategized to support the development of the financial sector. On the other hand, governments should increase their spending on productive sectors and promote inclusive growth through sustainable policy initiatives—for example, barriers to employment should be removed to encourage businesses to create more jobs. However, restrictions on capital investment inflows and high tax rates should be reduced to ensure inclusive and sustainable financial development. Our findings set out the following future research directions:

1. Identify the mechanisms through which tax burden and investment freedom affect financial development.

2. Investigate the impact on other policy variables on financial development.

3. Analyse the effect of financial development on economic growth and possible transmission channels.

4. Consider country-specific variations that could identify the idiosyncratic effect of tax burden, investment freedom and financial development

## Supporting information

**S1 File. This is the file that contains S1 to S6 Tables and Appendix.**
(PDF)

## Author Contributions

**Conceptualization:** Zhengrong Yang, Prince Asare Vitenu-Sackey.

**Formal analysis:** Prince Asare Vitenu-Sackey, Lizhong Hao.

**Funding acquisition:** Zhengrong Yang.

**Investigation:** Zhengrong Yang.

**Methodology:** Prince Asare Vitenu-Sackey, Yuqi Tao.

**Supervision:** Lizhong Hao.

**Writing – original draft:** Prince Asare Vitenu-Sackey, Lizhong Hao.

**Writing – review & editing:** Prince Asare Vitenu-Sackey, Yuqi Tao.

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
