## [Decision Letter · Decision Letter 0]

13 Apr 2023

PONE-D-23-00610Economic freedom, inclusive growth, and financial development: A heterogeneous panel analysis of developing countriesPLOS ONE

Dear Dr. Vitenu-Sackey,

Thank you for submitting your manuscript to PLOS ONE. After careful consideration, we feel that it has merit but does not fully meet PLOS ONE’s publication criteria as it currently stands. Therefore, we invite you to submit a revised version of the manuscript that addresses the points raised during the review process.

We look forward to receiving your revised manuscript.

Kind regards,

Nikeel Nishkar Kumar

Academic Editor

PLOS ONE

Journal Requirements:

Additional Editor Comments:

Please address the outstanding comments from Reviewer 1.

Reviewers' comments:

Reviewer's Responses to Questions

**Comments to the Author**

1. Is the manuscript technically sound, and do the data support the conclusions?

Reviewer #1: Yes

Reviewer #2: Yes

2. Has the statistical analysis been performed appropriately and rigorously? 

Reviewer #1: I Don't Know

Reviewer #2: Yes

3. Have the authors made all data underlying the findings in their manuscript fully available?

Reviewer #1: Yes

Reviewer #2: Yes

4. Is the manuscript presented in an intelligible fashion and written in standard English?

Reviewer #1: Yes

Reviewer #2: Yes

5. Review Comments to the Author

Reviewer #1: 1. Introduction – 3rd sentence. Reference the sentence.

2. The explanation for table 4 should be placed after table 4. This will indicate a better flow of the paper.

3. The paper looks very lengthy. Some tables from the results and discussion section can be submitted as supplementary material.

4. What was the justification for selecting these specific 72 countries? Countries included are from different regions, sizes, levels of development among many other differences.

5. How do the findings of your study set out for future research in this area?

Reviewer #2: I would like to express my appreciation to the authors for submitting their study in the knowledge of economic development.

The titled paper, "Economic freedom, inclusive growth, and financial development: A heterogeneous panel analysis of developing countries," has the potential to be published. However, I have a few suggestions for further improvements to the manuscript. Firstly, it would be beneficial if the authors adhere to strict guidelines for in-text referencing, which can be found in the referencing guidelines. Secondly, I recommend that the authors revisit the contributions of their study in the conclusion section and establish a connection with the specific theory being used. Lastly, since there is a limited amount of literature published in PLOS ONE as relative to the present research , I kindly request that the authors use relevant papers as a guide for their study.

I wish the authors all the best in editing their paper and hope they will take these points into consideration.

6. PLOS authors have the option to publish the peer review history of their article (what does this mean?). If published, this will include your full peer review and any attached files.

Reviewer #1: No

Reviewer #2: No

---

## [Author Response · Author response to Decision Letter 0]

26 Apr 2023

REVIEWER ONE 

1. Introduction – 3rd sentence. Reference the sentence. 

The sentence has been referenced accordingly.

2. The explanation for table 4 should be placed after table 4. This will indicate a better flow of the paper. 

Table 4 has been placed under the heading panel unit root and cross-sectional dependence tests which correspond to the explanation.

3. The paper looks very lengthy. Some tables from the results and discussion section can be submitted as supplementary material. 

Tables 8, 9, 10, 11 together with the appendix information have all been included attached supplementary results. 

4. What was the justification for selecting these specific 72 countries? Countries included are from different regions, sizes, levels of development among many other differences. The countries were selected based on the IMF's financial development index, with only those scoring below 0.5 out of 1 being considered.

5. How do the findings of your study set out for future research in this area? 

Our findings set out the following future research directions:

1. Identify the mechanisms through which tax burden and investment freedom affect financial development.

2. Investigate the impact on other policy variables on financial development.

3. Analyse the effect of financial development on economic growth and possible transmission channels.

4. Consider country-specific variations that could identify the idiosyncratic effect of tax burden, investment freedom and financial development

REVIEWER TWO 

1. it would be beneficial if the authors adhere to strict guidelines for in-text referencing, which can be found in the referencing guidelines. 

In-text references have been rectified. 

2. I recommend that the authors revisit the contributions of their study in the conclusion section and establish a connection with the specific theory being used. 

A connection between the theoretical position of the study and our findings have been established and highlighted in the conclusion. 

3. since there is a limited amount of literature published in PLOS ONE as relative to the present research, I kindly request that the authors use relevant papers as a guide for their study. 

1. Ding G, Vitenu-Sackey PA, Chen W, Shi X, Yan J, Yuan S. The role of foreign capital and economic freedom in sustainable food production: Evidence from DLD countries. Plos one. 2021 Jul 26;16(7):e0255186

2. Huang J, Ulanowicz RE. Ecological network analysis for economic systems: growth and development and implications for sustainable development. PloS one. 2014 Jun 30;9(6):e100923.

3. Singhal A, Sahu S, Chattopadhyay S, Mukherjee A, Bhanja SN. Using night time lights to find regional inequality in India and its relationship with economic development. PloS one. 2020 Nov 16;15(11):e0241907.

---

## [Decision Letter · Decision Letter 1]

26 Jun 2023

Economic freedom, inclusive growth, and financial development: A heterogeneous panel analysis of developing countries

PONE-D-23-00610R1

Dear Dr. Prince,

We’re pleased to inform you that your manuscript has been judged scientifically suitable for publication and will be formally accepted for publication once it meets all outstanding technical requirements.

Kind regards,

Nikeel Nishkar Kumar

Academic Editor

PLOS ONE

Additional Editor Comments (optional):

Reviewers' comments:

Reviewer's Responses to Questions

**Comments to the Author**

1. If the authors have adequately addressed your comments raised in a previous round of review and you feel that this manuscript is now acceptable for publication, you may indicate that here to bypass the “Comments to the Author” section, enter your conflict of interest statement in the “Confidential to Editor” section, and submit your "Accept" recommendation.

Reviewer #1: All comments have been addressed

Reviewer #2: All comments have been addressed

2. Is the manuscript technically sound, and do the data support the conclusions?

Reviewer #1: Yes

Reviewer #2: Yes

3. Has the statistical analysis been performed appropriately and rigorously? 

Reviewer #1: I Don't Know

Reviewer #2: Yes

4. Have the authors made all data underlying the findings in their manuscript fully available?

Reviewer #1: (No Response)

Reviewer #2: Yes

5. Is the manuscript presented in an intelligible fashion and written in standard English?

Reviewer #1: Yes

Reviewer #2: Yes

6. Review Comments to the Author

Reviewer #1: (No Response)

Reviewer #2: (No Response)

7. PLOS authors have the option to publish the peer review history of their article (what does this mean?). If published, this will include your full peer review and any attached files.

Reviewer #1: No

Reviewer #2: No

---

## [Editor Report · Acceptance letter]

29 Jun 2023

PONE-D-23-00610R1 

Economic freedom, inclusive growth, and financial development: A heterogeneous panel analysis of developing countries 

Dear Dr. Vitenu-Sackey:

I'm pleased to inform you that your manuscript has been deemed suitable for publication in PLOS ONE. Congratulations! Your manuscript is now with our production department. 

Kind regards, 

on behalf of

Dr. Nikeel Nishkar Kumar 

Academic Editor

PLOS ONE